# Etiological Aspects of Infectious Endocarditis in a Tertiary Hospital in Northeastern Romania

**DOI:** 10.3390/medicina61010095

**Published:** 2025-01-09

**Authors:** Isabela Ioana Loghin, Amelia Elena Surdu, Șerban Alin Rusu, Ion Cecan, Victor Daniel Dorobăț, Amelia Andreea Mihăescu, Carmen Mihaela Dorobăţ

**Affiliations:** 1Department of Infectious Diseases, “Grigore T. Popa” University of Medicine and Pharmacy, 700115 lasi, Romania; 2Department of Infectious Diseases, “St. Parascheva” Clinical Hospital of Infectious Diseases, 700116 Iasi, Romania; rususerbanalin@yahoo.com (Ș.A.R.); ion11cecan@gmail.com (I.C.); ameliamihaescu@gmail.com (A.A.M.); carmendorobat@yahoo.com (C.M.D.); 3Department of Implantology, Removable Prostheses and Technology, Faculty of Dental Medicine, “Grigore T. Popa” University of Medicine and Pharmacy, 700115 Iasi, Romania; 4Department of Intensive Care, University Hospital of Emergency, 050098 Bucharest, Romania; victordorobat@yahoo.com

**Keywords:** infective endocarditis, etiological agents, antibiotics, duke criteria, echocardiography

## Abstract

*Background and Objectives*: Infective endocarditis (IE) is a severe, life-threatening infection of the endocardial surface. Its incidence has shifted towards older, immunocompromised patients and those with cardiac devices. *Materials and Methods*: This study was conducted at the “Sf. Parascheva” Clinical Hospital of Infectious Diseases in Iasi, Romania, and retrospectively analyzed cases of IE from January 1, 2019, to September 30, 2024. It received ethical approval (Approval No. 7/17 June 2024). *Results*: The study included 130 patients with infectious endocarditis, predominantly men (75.38%), with a median age of 55 years. The most affected age groups were 50–59 and over 60 years, each representing 30.76% of cases. The most frequently implicated etiological agent was *Staphylococcus aureus* in 33% of cases. The most common antibiotic regimen combined glycopeptides and fluoroquinolones/polymyxins (27% cases). *Conclusions*: A multidisciplinary approach involving infectious disease specialists, cardiologists, and cardiovascular surgeons is essential for effective treatment. Immediate combined antibiotic therapy is vital for presumed IE cases. Despite advances in diagnosis and treatment, the high mortality rates highlight the importance of timely intervention. Future research should focus on improving preventive and therapeutic strategies for IE.

## 1. Introduction

Infective endocarditis (IE) is defined as a microbial infection of the endocardial surface. It represents “infection of the endocardium with the presence of microorganisms at the level of the lesion”, specifying that it may be in the valvular endocardium, parietal or septal defects, or a persistent arterial channel. This descriptive formulation emphasizes the initial endocardial changes, which are more common in approaches to the condition from a cardiological perspective [1,2].

In the last 50 years, there have been major changes in IE. Thus, a significant decrease in mortality was observed, the etiological spectrum changed, new antibiotics and new laboratory techniques appeared, echocardiography was developed, and the susceptible population changed, increasing the proportion of immunocompromised people and those with valve prostheses or pacemakers at the expense of those with rheumatic valvular diseases [3,4].

Over the last decades, shifts in etiology and epidemiology have been observed. The accurate incidence of IE is difficult to establish due to the variability in diagnostic criteria and reporting methodologies. IE continues to be a rare condition in Western countries, with its incidence influenced by geographic factors, predisposing risk factors, and the presence of underlying health conditions. Early diagnosis and appropriate initial management of IE remain particularly challenging, especially in the emergency department setting. Although the overall annual incidence of IE remained relatively stable, from 9 to 15 cases per 100,000 per year, this condition has progressively become more prevalent among older adults, with a significant increase in incidence observed in patients over the age of 60 and those with multiple comorbidities. It is important to note that the incidence of IE is higher in men than in women, with male-to-female ratios ranging. The incidence rate is greater in older individuals and shows an upward trend over time. In the younger population, congenital heart disease continues to be the predominant risk factor for IE, whereas intravenous drug use has emerged as a growing concern [5,6].

Community-acquired native valve IE is attributable to staphylococci, streptococci, or enterococci in approximately 90% of cases. These microorganisms are commensal species, commonly residing in the skin, oropharynx, and urogenital and gastrointestinal tracts [7,8].

IE involves a variety of etiological agents, which can vary depending on specific clinical scenarios. Acute IE is most commonly caused by *Staphylococcus aureus* (meticillin-sensitive *Staphylococcus aureus*—MSSA—or meticillin-resistant *Staphylococcus aureus*—MRSA), *Streptococcus pneumoniae,* and *Haemophilus influenzae*. Subacute IE, on the other hand, is typically associated with *Streptococcus viridans*, *Enterococcus* spp., and *Staphylococcus aureus*. Among intravenous drug users, the most frequently implicated pathogens include *Staphylococcus aureus*, *Pseudomonas aeruginosa*, *Candida* spp., and *Enterococcus* spp. In the case of the prosthetic valve, the primary causative agents of IE are *Staphylococcus epidermidis*, *Staphylococcus aureus*, and group B streptococci [9,10].

The main risk factors associated with IE are poor dental hygiene, diabetes mellitus, immunosuppression, systemic sepsis, and recent surgical or non-surgical invasive procedures [11,12]. The clinical diagnosis involves a combination of major criteria (positive blood cultures, evidence of valvular damage in the echocardiographic examination) and minor criteria (fever, manifestations of glomerulonephritis, Osler nodules, predisposing cardiac factors, suggestive but not conclusive echocardiographic images to be considered a major criterion, splenomegaly, elevated erythrocyte sedimentation rate (ESR), elevated C-Reactive Protein (CRP), presence of venous catheters, microscopic hematuria).

Bacteriological diagnosis consists of highlighting the etiological agent in blood cultures (three samples are recommended), and dynamic transthoracic and/or transesophageal echocardiography allows the definite confirmation of valvular changes [13,14].

Given that IE continues to be a significant concern and the associated risk factors are becoming increasingly prevalent, it is essential to adopt a comprehensive approach that emphasizes commonly encountered aspects. This will facilitate enhanced understanding and enable prompt intervention [15,16].

The goal of this study is to illustrate the traits and development of IE patients. Also, we aim to highlight the etiology of infective endocarditis in the northeastern region of Romania, along with the evaluation skills needed to evaluate possible cases, the available treatment and management options for IE in our country, and the interprofessional team strategies for enhancing communication and care coordination to enable better outcomes and more efficient care for IE patients.

## 2. Materials and Methods

### 2.1. Database Description

To highlight the characteristics and related comorbidities of IE cases, we conducted a retrospective clinical study on patients who were hospitalized at the “Sf. Parascheva” Clinical Hospital of Infectious Diseases from Iasi in the northeastern region of Romania. The time frame under investigation was 1 January 2019, to 30 September 2024.

### 2.2. Ethical Approval

The “St. Parascheva” Clinical Hospital of Infectious Diseases in Iasi, Romania, gave the study its clearance (17 June 2024; Approval No. 7/2024). Written informed consent was obtained from the patients when they were admitted to our hospital, according to the hospital’s policy.

### 2.3. Study Design

Data on demographics specific to age and gender, individual pathological histories, clinical characteristics, blood tests, assessments of possible bacteriological infections, patient comorbidities, the initiation of antibiotic therapy, and the course and prognosis of infectious endocarditis patients were all collected. Modified Duke criteria were used for the diagnosis of IE. Further, the presence of 2 major criteria, 1 major and 3 minor criteria, or 5 minor criteria suggests definite IE, while the presence of 1 major and 1 minor or 3 minor criteria suggests possible IE.

### 2.4. Study Setting

The “St. Parascheva” Clinical Hospital of Infectious Disease Iasi is a primary referral medical facility in the northeastern region of Romania, with a capacity of 300 beds. It is divided into six pavilions. The enrolled patients were evaluated by echocardiographic imaging (transthoracic echocardiography and transesophageal echocardiography). Contact was maintained with the cardiologist and cardiovascular surgeon regarding the evolution of the cases.

The “St. Parascheva” Clinical Hospital of Infectious Disease Iasi laboratory benefits from accreditation by the Romanian Accreditation Association with ISO 15189/2013 standards, an obligation that is unique in Europe [17].

All blood tests were performed by the hospital’s central laboratory. Blood cultures were collected and antibiograms were made after the etiologic agent was found.

### 2.5. Statistical Analysis

The Pearson test in the XLSTAT version 2019 program was used to determine the correlation between demographic characteristics, clinical data, and the results. Kendall’s Tau correlation coefficients were established. Statistical Software for Excel (XLSTAT 2019.4) version 2019 was used to conduct the statistical analysis. The descriptive data are displayed as means, percentages, and absolute values. The χ2 and unpaired Student *t*-tests were used to determine whether the differences between the groups were statistically significant.

## 3. Results

In our clinical hospital in the northeastern part of Romania, 130 patients diagnosed with IE were recorded from 1 January 2019, to 30 September 2024.

Men most frequently presented with infectious endocarditis (98 cases, 76%) compared with women (32 cases, 24%). Most of the instances involved adults aged between 50 and 59—40 patients (31%)—and over 60—40 cases (31%). Next were the age groups 40–49 years—30 patients (23%); 30–39 years—15 patients (11%); and 20–29 years—5 patients (4%). The study group’s median age was 55 years old (Table 1).

Due to the SARS-CoV2 pandemic, we registered low admissions between 2020 and 2021, with 7 cases (5%) and 10 cases (8%), respectively. The year with the most admissions, in the study period, was 2019 with 45 cases (35%) (Table 2).

The portal of entry (POE) for the present IE cases was identified in 93 patients (71%). Among the identified POEs, 42% were cutaneous, 27% were oral or dental, and 23% were gastrointestinal (Figure 1).

At admission, the most frequent symptoms were cardiac murmurs, noted in 117 patients (90%), followed by fever and chills in 110 patients (85%), and physical and mental fatigue in 104 patients (80%). Dyspnea on minimal exertion was reported in 90 patients (69%), while precordial pain was present in 52 patients (40%). Neurological symptoms, such as obnubilation, paresis, and paresthesias, were rare, affecting only five patients (4%). These data highlight cardiac murmurs and fever as key clinical indicators for diagnosis (Table 3).

Modified Duke criteria were used to diagnose or presumptively diagnose IE. Further, the presence of two major criteria, one major and three minor criteria, or five minor criteria suggests definite IE, while the presence of one major and one minor or three minor criteria suggests possible IE (Table 4).

The localization of vegetation detected through transthoracic or transesophageal echocardiography showed a higher frequency on the mitral valve, observed in 51% of cases, followed by the aortic and tricuspid valves, each with an incidence of 15% and 12%. Vegetations located on the pulmonary valve were less frequent (11%). There were 15 cases (12%) that had presented normal echocardiographic findings, but according to the modified Duke criteria were registered as possible IE (Table 5). Ten patients (8%) had prosthetic mitral valve implantation before the IE episode.

Cardiac IE-predisposing factors were found in 23% of patients (30 cases), 5% (6 cases) presented mitral valve prolapse, 10% (13 cases) had hypertrophic cardiomyopathy, and 8% (11 cases) had an anterior episode of endocarditis. The percentage of patients with primary IE was 77% (100 cases).

The characterizations of vegetation revealed that most were mobile, observed in 88% of cases. Regarding size, 81% of vegetation was smaller than 10 mm, while only 12% was classified as large (>10 mm). Ten cases presented multiple vegetations in various locations (7%) (Table 6).

Blood cultures were collected and an etiologic agent was found in 103 cases (84%). The most frequently implicated etiological agent was *Staphylococcus aureus* with 33% of total strains and identified in 43 of patients, followed by *Enterococcus faecalis* with 20% of total strains and identified in 26 patients; *Viridans Group Streptococci* were found in 15% of total strains and identified in 20 patients; and coagulase-negative *Staphylococcus epidermidis* was found in 10 patients (13%). It should be noted that the Gram-negative bacterium involved was *Klebsiella pneumoniae* in four cases (3.07%), (Table 7).

Broad-spectrum antibiotic therapy for which the antibiogram showed sensitivity to beta-lactams, cephalosporins, glycopeptides, trimethoprim-sulfamethoxazole, rifampicin, quinolones, lincosamides, oxazolidinones, and thus specific treatment was initiated according to the antibiogram.

The therapeutic regimens were with glycopeptide + fluoroquinolone/polymyxin (27%), aminopenicillins and quinolones administered (19%), third-generation cephalosporins + quinolones/aminopenicillins (15%), oxazolidinone + quinolones (8%), carbapenem + quinolones (6%), and lincosamide + quinolone (4%) (Table 8).

For the cases with negative blood cultures, treatment was initiated with aminopenicillins/aminoglycosides + fluoroquinolons (14 cases, 11%) and glycopeptides + fluoroquinolone/polymyxin (13 cases, 10%), with positive clinical and biological response.

The treatment instituted led to negative blood cultures in the first 4–7 days, but an extension of antibiotic therapy was necessary to sterilize the vegetation present on echocardiography.

Between the years of the SARS-CoV2 pandemic (2020 and 2021), there were 17 cases (13%) recorded, from which 7 cases (6%) presented SARS-CoV2 coinfection (4 cases (3%) were male and 3 (2%) were female). Regardless of the small number of SARS-CoV2 coinfection cases during this period, the identified spectrum was the same, with positive hemocultures for *Staphylococcus aureus* (three cases, 2%), *Viridans Group Streptococci* (two cases, 1%) and *Enterococcus faecalis* (two cases, 1%). Broad-spectrum antibiotic therapy for which the antibiogram showed sensitivity to glycopeptide + fluoroquinolone/polymyxin (three cases, 2%) and aminopenicillin/aminoglycoside + fluoroquinolone (four cases, 3%) was administered. Also, the imaging criteria were met—echocardiography was performed for these cases and four cases (3%) presented normal cardiac valves, while in the other three cases (2%), mobile and < 10 mm vegetations were recorded.

Pathogenic and antithrombotic treatment was used (low-molecular-weight heparin, calculated per kg) in collaboration and recommended by the cardiologist, with favorable outcomes.

In the studied cases, a favorable evolution was most frequently observed in 58 cases (45%). A reserved prognosis was noted in 13% of the investigated patients, with the registration of death in 5% (seven cases). The share of patients with endocarditis transferred for cardiological consultation was 28 (22%). Patients were evaluated by the cardiovascular surgeon, and 20 (15%) of the subjects were transferred to cardiovascular surgery services for intervention (following the cardiovascular surgeon’s recommendation) (Table 9).

Major complications included heart failure (seven cases, 5%), pericarditis (three cases, 2%), and thromboembolic events (three cases, 2%). Although their incidence was low, these complications significantly contributed to increased mortality and morbidity (Table 10).

## 4. Discussion

Rajani R. et al. stated that despite advancements in microbiological and diagnostic methods, infectious endocarditis is linked to a high rate of morbidity and mortality. Prompt surgical intervention when necessary and early diagnoses with the early inclusion of a specialized IE team are proven strategies that enhance patient outcomes. To provide fair IE care across various clinical networks, further patient pathway development is needed [18].

Holland T.L. et al. concluded that imaging technology will continue to advance, and further research is needed to define which patients with suspected IE should undergo transoesophageal echocardiography and which patients may benefit from newer imaging modalities. Novel Gram-positive antibiotics are promising but as yet untested in IE. If proven to be effective, they might enable simpler and more patient-friendly treatment regimens. The debate around IE prophylaxis will likely continue until prophylaxis strategies are compared prospectively. Vaccine development has not yet yielded an effective and commercially available product, but numerous candidates are in the pipeline [19].

Since infectious endocarditis is a complicated condition, experts from a variety of disciplines must contribute. In both IE research and disease management, it is imperative to prioritize a multidisciplinary approach. Even though IE has been diagnosed and treated more effectively in recent decades, there are still many unsolved issues, and randomized clinical studies are desperately needed to learn more about this difficult illness. There is potential for better patient outcomes thanks to advancements in AI-powered machine learning and imaging technologies [20].

Endocarditis is becoming more common as the population ages, invasive medical device use rises, and the opioid crisis persists. The illness is still deadly and challenging to identify. Thankfully, we are still learning more about the illness. New imaging and immunoassay methods are available to help with diagnosis. The recently updated modified Duke criteria are still applicable. The guidelines for antibiotic treatment are always changing, and newer drugs—particularly dalbavancin—have the potential to enhance treatment. All healthcare professionals should be aware of the continuous modifications to our strategy for treating infective endocarditis [21].

Lindberg H. et al. observed that in 4050 episodes of bacteremia, the modified Duke criteria assigned 307 episodes (7.6%) as confirmed IE, 1190 (29%) as possible IE, and 2553 (63%) as rejected IE. Using the Duke-ISCVID criteria, 13 episodes (0.3%) were reclassified from possible to definite IE, and 475 episodes (12%) were reclassified from rejected to possible IE. With the modified Duke criteria, 79 episodes that were treated as IE were classified as possible IE, and 11 of these episodes were reclassified to definite IE with Duke-ISCVID. Applying the decision to treat IE as a reference standard, the sensitivity of the Duke criteria was 80% [22].

One limitation of our study is that we did not record rare etiologic agents, particularly members of the HACEK group. This is probably due to the limited possibilities of the bacteriological laboratory of the hospital, which uses hemocultures tests.

## 5. Conclusions

The co-occurrence of bacteremia and a murmur in a feverish patient strongly suggests the possible presence of IE from the standpoint of solely clinical signs. To properly evaluate the type of valvular lesion, identify vegetation, and determine the severity of any problems related to IE, an echocardiography must be performed as soon as feasible. It is wise to start with transthoracic echocardiography (TTE) because of its non-invasiveness, simplicity, and superior myocardial function information (such as ejection fraction), even though transesophageal echocardiography (TEE) has a higher sensitivity for detecting valvular vegetation and paravalvular problems. Given the high likelihood that the patient may have IE, a TEE should be performed if a TTE is determined to be negative or inconclusive [23].

The most frequently implicated etiological agents found in blood cultures were *Staphylococcus aureus*, followed by *Enterococcus, Viridans Group Streptococci*, and *Staphylococcus epidermidis*. The Gram-negative bacteria involved *Klebsiella pneumoniae*. We did not record rare etiologic agents, particularly members of the HACEK group. This is probably due to the limited possibilities of the bacteriological laboratory of the hospital, which uses hemoculture tests.

According to the antibiogram, a particular treatment was started for broad-spectrum antibiotic therapy with beta-lactams, cephalosporins, glycopeptides, trimethoprim-sulfamethoxazole, rifampicin, quinolones, lincosamides, and oxazolidinones.

To deliver care, a multidisciplinary team would be assembled, comprising experts in infectious disease, cardiology, and cardiovascular surgery. To treat presumed IE, combined antibiotic therapy must be administered as soon as possible.

Despite advancements in microbiological and diagnostic methods, IE is linked to a high rate of morbidity and mortality in our country. Prompt surgical intervention when necessary and early diagnosis with the early inclusion of a specialized IE team are proven strategies that enhance patient outcomes. To provide fair IE care across various clinical networks, further patient pathway development is needed. When treating feverish patients, clinicians should be on the lookout for signs including bacteremia and heart murmurs.

In conclusion, we are emphasizing the necessity of early diagnosis and vigorous treatment to address the ongoing high fatality rates of IE. When treating feverish patients, clinicians should be on the lookout for signs including bacteremia and heart murmurs. Future studies should concentrate on creating preventative measures and treatment plans that are more successful.

## Figures and Tables

**Figure 1 medicina-61-00095-f001:**
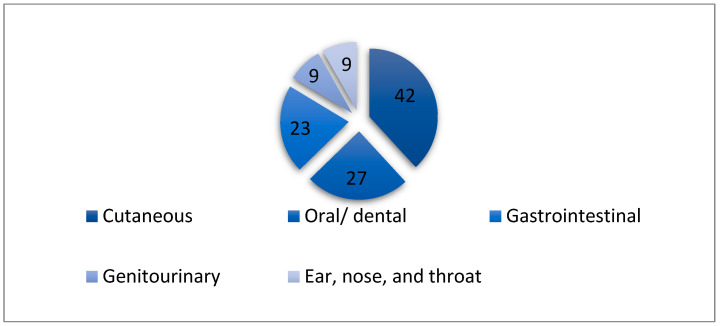
Portal of entry for the present IE cases.

**Table 1 medicina-61-00095-t001:** Distribution of the study cases according to age and gender.

Age (Years)	Male (*n*, %)	Female (*n*, %)	Total Number of Cases (*n*, %)
20–29	3 (2%)	2 (2%)	5 (4%)
30–39	10 (7%)	5 (4%)	15 (11%)
40–49	20 (16%)	10 (7%)	30 (23%)
50–59	30 (24%)	10 (7%)	40 (31%)
Over 60	35 (27%)	5 (4%)	40 (31%)
Total	98 (76%)	32 (24%)	130 (100%)

**Table 2 medicina-61-00095-t002:** Distribution of the cases according to the year of hospitalization and gender.

Hospitalization Year	Number of Cases (*n*, %)	Male (*n*, %)	Female (*n*, %)
2019	45 (35%)	35 (28%)	10 (7%)
2020	7 (5%)	5 (4%)	2 (1%)
2021	10 (8%)	7 (5%)	4 (3%)
2022	21 (16%)	18 (14%)	3 (2%)
2023	25 (19%)	17 (13%)	7 (6%)
2024	22 (17%)	16 (12%)	6 (5%)

**Table 3 medicina-61-00095-t003:** Distribution of the cases according to the symptoms at admission.

Clinical Symptoms and Signs	Number of Cases (%)
cardiac murmurs	117 (90%)
physical and mental asthenia	104 (80%)
fever, chills	110 (85%)
dyspnea on light exertion	90 (69%)
chest pain	52 (40%)
cough	39 (30%)
headache	45 (35%)
numbness, paresis, paresthesias	5 (4%)

**Table 4 medicina-61-00095-t004:** Applied Duke criteria for the study cases.

Modified Duke Criteria	Number of Cases (*n*, %)
Definite IE - 2 major criteria - 1 major criterion + 3 minor criteria - 5 minor criteria	80 cases, 61% 20 cases, 15% 3 cases, 4%
Possible IE - 1 major criterion + 1 minor criterion - 3 minor criteria	12 cases, 9% 15 cases, 11%

**Table 5 medicina-61-00095-t005:** Echocardiographic distribution of the valve vegetation of the study patients.

Echocardiographic Locations of Vegetation	Number of Cases (%)
Tricuspid valve	15 (12%)
Aortic valve	20 (15%)
Mitral valve	66 (51%)
Pulmonary valve	14 (11%)

**Table 6 medicina-61-00095-t006:** The characteristics of the vegetation are related to the number of cases.

Vegetation Character	Number of Cases (%)
Mobile vegetation	114 (88%)
Vegetation < 10 mm	105 (81%)
Large vegetation (>10 mm)	15 (12%)
Multiple vegetations	10 (7%)

**Table 7 medicina-61-00095-t007:** Etiology spectrum in study patients with infectious endocarditis.

Etiologic Agent	Number of Cases (*n*, %)
*Staphylococcus aureus*	43 cases, 33%
*Enterococcus faecalis*	26 cases, 20%
*Viridans Group Streptococci*	20 cases, 15%
*Staphylococcus epidermidis*	10 cases, 13%
*Klebsiella pneumoniae*	4 cases, 3%

**Table 8 medicina-61-00095-t008:** Comparative analysis of therapeutic regimens.

Therapeutic Regimen	Number of Cases (*n*, %)
Glycopeptide + fluoroquinolone/polymyxin	35 (27%)
Aminopenicillin/aminoglycoside + fluoroquinolone	25 (19%)
Third-generation cephalosporins + fluoroquinolone/aminoglycoside	20 (15%)
Oxazolidinone + quinolone	10 (8%)
Carbapenem + quinolone	8 (6%)
Lincosamide + quinolone	5 (4%)

**Table 9 medicina-61-00095-t009:** Prognosis of the study cases.

Evolution	Number of Cases (*n*, %)
Favorable	58 (45%)
Reserved	17 (13%)
Deceased	7 (5%)
Transfer of cardiovascular surgery services	20 (15%)
Cardiology transfer	28 (22%)

**Table 10 medicina-61-00095-t010:** Complications during the evolution of the patients.

Major Complications	Number of Cases (*n*, %)
Heart failure	7 (5%)
Pericarditis	3 (2%)
Thromboembolic events	3 (2%)

## Data Availability

All data generated or analyzed during this study are included in this published article.

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
