# Peer review of "Etiological Aspects of Infectious Endocarditis in a Tertiary Hospital in Northeastern Romania"

_medicina, 2025, doi:10.3390/medicina61010095_

Round 1
Reviewer 1 Report
Comments and Suggestions for Authors
See file, lease.

Reviewer 2 Report
Comments and Suggestions for Authors
The topic of this peer-reviewed manuscript is interesting and of potential value to “Medicina” readers.
Article „Etiological Aspects of Infectious Endocarditis in a Tertiary Hospital from North Eastern Romania” by Isabela Ioana Loghin et al. is a retrospective study that analyzed 130 cases of infective endocarditis (IE) conducted at the tertiary hospital "Sf. Parascheva" Clinical Hospital of Infectious Diseases in a North Eastern Romania, Iasi, and presented etiological aspects of IE.
1. Typo error: %) in the abstract.
2. Write percentages uniformly, without decimal places or with 1-2.
3. Use the unique style for preparing the references.
4. I could not find references 5 and 6?
5. Abbreviations should be defined the first time they appear in each of three sections: the abstract; the main text; the first figure or table (MSSA, MRSA, POE).
6. I suggest adding a row in Table 1 with number of male/female patients (in total), presenting number and percentage.
7. There is no distribution of the cases according to the gender presented in Table 2?
8. “Regarding size, 81.25% of vegetation were smaller than 10 mm, while only 12.5% were classified as large (>10 mm). One case presented multiple vegetations in various locations (6.25%)”, that is 121 in total; the remaining 9 patients are lacking this information?
9. The text in the manuscript does not correspond to the information in Table 8?
10. All figures and tables should be inserted into the main text close to their first citation.
11. Rearrange Table 9; put the number of cases (n, %) and the percentages in the same column.
12. Number of cases with major complications is not the same in the main text and table 10.
13. Which one of these statements is correct? “This was a retrospective study, and written informed consent had been obtained from the patients when they were admitted to our hospital, according to the hospital policy.” OR “At admission, every individual signed a waiver of informed consent, included in the admission protocol.”
Minor revision of the English language required.
Round 2
Reviewer 1 Report
Comments and Suggestions for Authors
Minor editorial comments.
After the abbreviation IE is introduced, the full term “infective endocarditis” continues to be used (lines 76, 88, 97, 115 142 and others).
The objective is too general, it is recommended to make it more specific.
The “statistical analysis” section should include a brief description of all methods that were used in the study.
The “results” section still lacks an analysis of prior heart disease that may have been the background for the secondary IE. The percentage of patients with primary IE is not specified.
The conclusion is also too global in content and does not reflect the findings of the study.
Author Response
Dear Editors and Referees,
First of all, we would like to express our thanks, the comments regarding the manuscript were very useful, helping to enrich the transmission of information and add value to our research.
According to the reviewer's suggestions, we added more data to support our observational retrospective research and express more clearly our goal and results obtained in North-Eastern Romania regarding the etiological Aspects of Infectious Endocarditis.
We adjusted the body text of the manuscript, with modifications in all the sections: introduction, material and methods, results, and conclusion.
We modified the typo error regarding the “infective endocarditis” abbreviations.
We clarified and developed the description of our study objective.
We added more data regarding the “Statistical Analysis” section.
We added in the "Results" section an analysis of prior heart disease that may have been the
background for the secondary IE, and the percentage of patients with primary IE.
We modified the conclusion so that it will follow the results.
